# Nutraceutical Strategies for Targeting Mitochondrial Dysfunction in Neurodegenerative Diseases

**DOI:** 10.3390/foods14132193

**Published:** 2025-06-23

**Authors:** Federica Davì, Antonella Iaconis, Marika Cordaro, Rosanna Di Paola, Roberta Fusco

**Affiliations:** 1Department of Chemical, Biological, Pharmaceutical and Environmental Sciences, University of Messina, Viale F. Stagno d’Alcontres 31, 98166 Messina, Italy; federica.davi@studenti.unime.it (F.D.); aiaconis@unime.it (A.I.); rfusco@unime.it (R.F.); 2Department of Biomedical, Dental and Morphological and Functional Imaging, University of Messina, Via Consolare Valeria, 98125 Messina, Italy; marika.cordaro@unime.it; 3Department of Veterinary Sciences, University of Messina, Viale SS Annunziata, 98168 Messina, Italy

**Keywords:** mitochondrial dysfunction, oxidative stress, antioxidants, neurodegenerative diseases, nutraceutical

## Abstract

In neurons, mitochondria generate energy through ATP production, thereby sustaining the high energy demands of the central nervous system (CNS). Mitochondrial dysfunction within the CNS was implicated in the pathogenesis and progression of neurodegenerative diseases, such as Alzheimer’s disease, Parkinson’s disease, amyotrophic lateral sclerosis, and multiple sclerosis, often involving altered mitochondrial dynamics like fragmentation and functional impairment. Accordingly, mitochondrial targeting represents an alternative therapeutic strategy for the treatment of these disorders. Current standard drug treatments present limitations due to adverse effects associated with their chronic use. Therefore, in recent years, nutraceuticals, natural compounds exhibiting diverse biological activities, have garnered significant attention for their potential to treat these diseases. It has been shown that these compounds represent safe and easily available sources for the development of innovative therapeutics, and by modulating mitochondrial function, nutraceuticals offer a promising approach to address neurodegenerative pathologies. We referred to approximately 200 articles published between 2020 and 2025, identified through a focused search across PubMed, Google Scholar, and Scopus using keywords such as “nutraceutical,” “mitochondrial dysfunction,” and “neurodegenerative diseases. The purpose of this review is to examine how mitochondrial dysfunction contributes to the genesis and progression of neurodegenerative diseases. Also, we discuss recent advances in mitochondrial targeting using nutraceuticals, focusing on their mechanisms of action related to mitochondrial biogenesis, fusion, fission, bioenergetics, oxidative stress, calcium homeostasis, membrane potential, and mitochondrial DNA stability.

## 1. Introduction

Neurodegenerative diseases (NDs) include a broad spectrum of neurological disorders characterized by progressive neuronal dysfunction within the central nervous system (CNS), culminating in neuronal cell death or degeneration [1,2,3], and ultimately leading to deficits in specific brain functions, such as movement, memory, and cognition. These pathological processes underlie the pathogenesis and progression of several NDs, including Alzheimer’s disease, Parkinson’s disease, amyotrophic lateral sclerosis, and multiple sclerosis [4,5].

The complex etiology of NDs involves several interconnected pathways, including mutated genes, abnormal protein accumulation, neuroinflammation, mitochondrial dysfunction, increased reactive oxygen species (ROS), apoptosis, endoplasmic reticulum (ER) stress, calcium overload, excitotoxicity, or neuronal death in specific brain regions [1,6].

Maintaining Ca^2+^ homeostasis across cellular, ER, and mitochondrial membranes is essential for diverse physiological functions. The disruption of these homeostatic mechanisms leads to persistent increases in intracellular and intra-organelle (ER and mitochondrial) Ca^2+^ concentrations, implicated in the pathogenesis of several neurodegenerative disorders [7,8]. Among the recognized mechanisms, mitochondrial dysfunction has garnered significant attention, as its disruption triggers neurotoxic pathways [9]. Indeed, mitochondria, often termed the “powerhouses” of the cell, play a critical role in NDs. These organelles generate cellular energy via ATP production, produce ROS through oxidative phosphorylation (OXPHOS), regulate calcium homeostasis, and initiate cell death pathways [10,11]. However, under conditions of stress, aging, or in the presence of NDs, neuronal mitochondria can undergo fragmentation and functional impairment. Mitochondrial dysfunction, marked by a loss of membrane potential, leads to both energy deficit and increased oxidative stress [12,13]. Mitochondria are highly dynamic organelles, constantly undergoing fusion and fission, processes that modulate their morphology and functionality. Mitochondrial dynamics, essential for proper cellular function, underpin numerous fundamental processes, including energy production, movement, differentiation, cell cycle progression, senescence, and apoptosis [14]. The dysregulation of these dynamics is a key pathogenic mechanism in various diseases and conditions characterized by mitochondrial dysfunction [15,16]. This is linked to NDs as it triggers the production of ROS, α-synuclein aggregation (which impairs mitochondrial transport, distribution, and morphology, thereby disrupting ATP delivery), decreased biogenesis, elevated mitophagy (the degradation of damaged mitochondria), and impaired fusion and fission, processes essential for maintaining the dynamics and function of mitochondria [17,18].

Recent studies have explored the growing interest in nutraceuticals as a complementary approach to standard drug therapy, targeting one or more of the above processes [5,19]. A nutraceutical can be defined as a product derived from food that offers health benefits beyond basic nutrition, and encompasses a wide range of natural products [20]. The increasing prevalence of this non-conventional approach stems from the diverse pathways or specific mechanisms of action through which nutraceuticals exert their effects in NDs. The potential benefits of nutraceuticals have led to their exploration as complementary therapies for the management of NDs [21,22]. These multifactorial agents can target various pathogenic mechanisms implicated in NDs, such as neuroinflammation, protein misfolding, ER stress, oxidative stress, and mitochondrial dysfunction. In many cases, the favorable safety profile of natural compounds, with fewer reported side effects, combined with the potential for synergistic effects when used in conjunction with conventional drug therapies, positions nutraceuticals as a promising integrative approach for improving the quality of life of patients with NDs [5,6].

This review examines the potential of various nutraceuticals, focusing on their diverse mechanisms of action within the complex pathophysiology of NDs and exploring the rationale for their use as complementary therapies. All the research was carried out across three major databases, i.e., PubMed, Google Scholar, and Scopus, focusing on studies published between 2020 and 2025, excluding earlier research to ensure up-to-date insights. The search strategy utilized both controlled vocabulary and free-text keywords, including “nutraceutical”, “mitochondrial dysfunction”, and “neurodegenerative diseases”. Initially, about 1200 articles based on the terms “nutraceuticals” and “neurodegenerative diseases” were selected for the chosen time period (2020–2025). In addition, 219 papers were considered by adding “mitochondrial dysfunction” as a keyword. From these, many were excluded because they were not reviews. This process produced a final selection of about 200 articles for study references.

## 2. Overview of Mitochondrial Biology and Dynamics

### 2.1. Structure and Function

Mitochondria are independent organelles, typically having an ellipsoid shape, characterized by a double-membrane structure with an inner membrane enclosing the mitochondrial matrix and an outer membrane surrounding the organelle [23].

The outer membrane (OMM) composition mirrors that of the cellular lipid membrane, enabling the diffusion of lipid-soluble molecules into the intermembrane space. The OMM, through various membrane complexes, allows the communication of mitochondria with cytosolic components; these include distinct forms of the translocase of the outer membrane (TOM) and mitochondrial import complex (TIM) [24,25,26].

The inner membrane (IMM) has a longer surface area than the OMM, which presents invaginations that give rise to characteristic cristae [25]. The respiratory chain complexes are situated in the IMM; thus, the cristae organization is important to raise the reaction surface. Moreover, cristae junctions are stabilized across proteins that are regulated by the Mitochondrial Contact Site and Cristae Organizing System (MICOS). It determines the shape of the cristae and, thus, the assembly and efficiency of the respiratory complexes; the defects in the arrangement of the cristae are directly associated with a deficit in the function of the mitochondria [27,28].

The OMM and IMM separate two aqueous compartments, the intermembrane space (IMS) and the inner most compartment, namely, the matrix. The IMS is crucial in various cellular processes, such as those related to respiration, metabolic support, and the translocation of proteins by identifying and transporting precursor proteins to the TIM through TIM chaperones (IMS-residing hexameric complexes) [24]. They are widely recognized for the ATP generation, whose success predominantly relies on the establishment of a chemical proton gradient [29]. The electron transport chain involves four multiprotein enzymatic complexes: complex I (NADH–ubiquinone oxidoreductase complex), complex II (succinate–coenzyme Q reductase complex), complex III (coenzyme Q–cytochrome c reductase), and complex IV (cytochrome c oxidase). These complexes utilize specific substrates, with complex I employing NADH, and complex II utilizing succinate and FADH2 [30].

Furthermore, mitochondria possess a singular characteristic unlike other cellular components: they harbor their own genome. This mitochondrial DNA (mtDNA) is a closed circular loop, double-stranded molecule of approximately 16,569 kilobases. This genome encodes 37 genes, which encode 13 proteins essential for the mitochondrial electron transport chain (e.g., subunits of NADH dehydrogenase 1, cytochrome b, cytochrome c oxidase I, and ATP synthase 6), two ribosomal RNAs, and 22 mitochondrial transfer RNAs crucial for the synthesis of mitochondrial proteins [31] (Figure 1).

### 2.2. Mitochondrial Dynamics

Mitochondria are highly dynamic organelles subject to constant processes, such as fission, fusion, mitophagy, and transport, ultimately determining their morphology, quality, quantity, distribution within cells, and the correct mitochondrial function. To allow the proper function of mitochondria, by means of mitochondrial dynamics, damaged components can be removed, or the compromised mitochondria themselves can be entirely eliminated through mitophagy, to avoid further cell damage [32]. To achieve optimal mitochondrial function and determine a cell’s fate, it is essential to maintain a balance of mitochondrial dynamics [33]. Mitochondrial fission is crucial for quality control as it enables the elimination of defective or non-functioning mitochondria and contributes to planned cell death in response to extreme cellular stress. Conversely, fusion enables the maintenance of overall function of mitochondria, as it allows mixing and exchange of intramitochondrial contents [32].

Furthermore, mitophagy is fundamental for quality control; by selectively eliminating damaged mitochondria, mitophagy prevents the release of pro-apoptotic proteins [34,35].

#### 2.2.1. Fission and Fusion

The processes of fusion and fission, which are an integral part of mitochondrial dynamics, are crucial for maintaining quality control in mitochondria [36,37,38]. They also play an important role in preserving mitochondrial growth, shape, distribution, and structure, which are fundamental to cellular homeostasis and survival [39,40]. These processes promote the separation of damaged mitochondria and the redistribution of their components.

Mitochondrial damage triggers the process of fission, which is necessary for the removal of these compromised structures. This process is mediated by a family of dynamin-related proteins (Drps), primarily including Drp1. The protein Drp1 is drawn from the cytosol to the mitochondrial membrane, where it interacts with receptors on the OMM, like mitochondrial fission factor (Mff) and fission 1 protein (Fis1). Subsequently, Drp1 oligomerizes with Drp2 to form a ring-like structure around the mitochondria, leading to membrane cleavage through GTP hydrolysis [41]. Therefore, Fis1, an inhibitor of the fusion machinery [33], plays a crucial role in driving mitochondrial fission by interacting with Drp1 and regulating the establishment of constriction sites at the ER–mitochondria interface [33,42]. Increased mitochondrial fission may result in smaller, potentially dysfunctional mitochondria, which may lack mitochondrial DNA (mtDNA) [43]. However, in potential degeneration, their smaller size enhances their transport through the intricate cytoskeletal network to distal cellular compartments [36,44].

Mitochondrial fusion is a process that involves the combination of two mitochondria, fusing their membranes and interchanging intracellular substances, including proteins, lipids, and metabolites. Maintaining the proper functioning of the electron transport chain (ETC) depends on this process, in particular on the distribution of complex I [45]. Three GTPases regulate fusion: mitofusin (Mfn) 1 and 2, situated on the OMM, along with optic atrophy 1 (Opa1) located on the IMM [46].

Mfn1 and 2 initiate the OMM fusion by forming homo- and hetero-oligomeric complexes. Subsequently, Opa1 activation facilitates the IMM fusion by creating a pore in the membrane [47,48].

Impaired fusion leads to increased mitochondrial fragmentation, the reduced expression of mitochondrial-encoded ETC proteins, and the inhibition of ATP synthesis, ultimately contributing to neuronal death. Therefore, while enhanced fusion preserves mitochondrial health in healthy cells, impaired fusion activity is linked to the onset of neurodegenerative diseases [49].

Given the high energy demands of neurons, the integrity of their mitochondrial network is paramount for proper neuronal function. Consequently, problems with mechanisms for maintaining mitochondria, such as imbalances between these two processes, can result in the buildup of dysfunctional mitochondria, which in turn can cause and worsen various CNS disorders.

#### 2.2.2. Mitophagy

Mitochondrial dysfunction is characterized by excessive calcium accumulation, the increased production of ROS, the opening of the mitochondrial permeability transition pore (mPTP), and the subsequent secretion of cytochrome c (cyt c), which can induce cell death by apoptosis [50].

Mitochondrial autophagy (mitophagy) is activated due to a loss of function, which is a selective degradation process. Consequently, this is essential for maintaining homeostasis, including mitochondrial biogenesis and the overall quality and quantity of mitochondria [51]. Mitophagy has been demonstrated to exert neuroprotective effects by eliminating dysfunctional mitochondria, consequently reducing ROS production [52]. The dynamic process involves two distinct phases. Initially, dysfunctional or damaged mitochondrial regions are recognized and enclosed within double-membraned autophagosomes. Subsequently, these autophagosomes merge with lysosomes, forming autolysosomes where the damaged mitochondria are broken down by hydrolytic enzymes. A well-characterized pathway of mitophagy involves the concerted action of PINK1 and Parkin. PINK1, a serine/threonine kinase, is a constitutively expressed protein on the OMM, and Parkin, an E3 ubiquitin ligase, can be phosphorylated at Ser65 in its UBL domain by PINK1 and recruited to damaged mitochondria in order to ubiquitinate its substrates [53]. In healthy mitochondria, PINK1 undergoes rapid proteolytic cleavage and subsequent degradation within the IMM. Mitochondrial damage causes IMM depolarization, which in turn hinders PINK1 degradation, allowing it to accumulate on the OMM. This accumulated PINK1 then recruits Parkin and activates its ubiquitin ligase activity through the phosphorylation of ubiquitin at Ser65.

Parkin-mediated ubiquitination enables the degradation of several OMM proteins, including Mfn1 and 2, and VDAC1, and simultaneously recruits autophagy receptors like p62 and optineurin (OPTN) [54]. Prior studies have revealed faulty mitophagy in NDs, resulting in the accumulation of autophagosomes, unusual endosomes, and lysosomes. This impairment negatively affects the mitophagic process, with a notably significant contribution of the advancement of neurodegenerative pathologies [55].

### 2.3. Mitochondrial Biogenesis

Mitochondrial biogenesis (MB) involves the growth and division of existing mitochondria to generate new ones, ultimately yielding a rise in the amount of mitochondria within the cell. This process, orchestrated by peroxisome proliferator-activated receptor-gamma (PPARγ) coactivator-1 alpha (PGC-1α), involves the coordinated synthesis of proteins encoded by both the nuclear and mitochondrial genomes, as far as the replication of mitochondrial DNA (mtDNA) [56].

PGC-1α acts as the primary controller of MB, integrating and coordinating the functions of several transcription factors, including nuclear respiratory factors 1 and 2 (NRF1-2) and mitochondrial transcription factor A (TFAM) [57]. NRF1-2 regulate the expression of numerous nuclear-encoded mitochondrial proteins, encompassing the components of the oxidative phosphorylation (OXPHOS) system, proteins involved in mitochondria import, and antioxidant defense proteins. Furthermore, NRF1-2 also activates mitochondrial gene transcription through interaction with the promoter sequences of TFAM [58,59]. In particular, PGC-1α recruits extra transcription factors to facilitate the regulation of nuclear encoded genes essential for MB. Its activation triggers the NRFs, which eventually leads to the transcription of genes linked to mitochondria, including ETC components like ATP synthase, cytochrome c, cytochrome oxidase IV, and TFAM. After translation, TFAM is relocated to the mitochondrial matrix where it aids in the replication of mtDNA and the expression of genes. PGC-1α also interacts with various transcription factors, including peroxisome proliferator-activated receptors (PPARs), glucocorticoid receptors, thyroid hormone receptors, estrogen receptors, antioxidant proteins, and mitochondrial transporters, further amplifying the regulatory network of MB [56].

## 3. Mitochondrial Dysfunction and Oxidative Stress in Common Neurodegenerative Disorders

As described above, mitochondria are key multifunctional organelles that perform many important functions in the cell. They are essential not only in energy production but also play a role in thermogenesis, calcium homeostasis, generation and maintenance of key cell metabolites, cell death, and redox signaling [60]. Mitochondrial dysfunction can result from various causes, such as mutations in mtDNA, a decrease in mitochondrial biogenesis, or changes in the processes of fusion and fission. When their function is compromised, they can generate even more ROS, thereby creating a vicious cycle of oxidative stress and mitochondrial malfunction [61]. The onset of several NDs is linked to mutations in mtDNA and respiratory chain issues [62]. Indeed, neurodegenerative disorders have several key characteristics in common, including abnormally accumulated aggregated proteins and the presence of oxidative damage and mitochondrial dysfunction. It is also known that all misfolded aggregate proteins, including β-amyloid, tau, and α-synuclein, can impede mitochondrial activity and trigger oxidative stress [63]. They are the very reasons why the central nervous system is particularly vulnerable to mitochondrial dysfunction, as these organelles play a crucial role in the construction and functioning of brain networks, so the breakdown of their homeostasis is often associated with progressive oxidative damage and general energy failure.

Mitochondrial homeostasis imbalance is present in several neurodegenerative diseases, such as Alzheimer’s disease, Parkinson’s disease, amyotrophic lateral sclerosis, and multiple sclerosis (Figure 2).

### 3.1. Alzheimer’s Disease

Alzheimer’s disease (AD) is a neurodegenerative disorder, characterized by a highly complex and varied pathophysiology involving the buildup of senile plaques formed by amyloid beta (Aβ) peptide deposited outside cells and the presence of intraneurofibrillary tangles (NFTs) produced from abnormally phosphorylated tau protein (pTau) [51]. The buildup of Aβ and pTau results in complex neuronal dysfunction that impacts synaptic signaling, mitochondrial function, neuroinflammation, and neuronal loss [64,65]. Research indicates that Aβ plaques reduce the accumulation of Ca^2+^ ions in the ER, resulting in an overload of Ca^2+^ in the cytosol, which subsequently leads to lower endogenous GSH levels and an increased accumulation of ROS [66]. Aβ and pTau aggregates within synaptic clefts, regions of high metabolic activity, thereby impairing synaptic mitochondrial function and neurotransmission [67]. Aβ accumulation within mitochondria compromises mitochondrial function, while tau protein disrupts mitochondrial transport in neurons, selectively disrupting ETC complex I, and further hindering mitochondrial respiration [51,68], thereby resulting in reduced ATP production and elevated oxidative stress. Severe oxidative stress, caused by a surge in ROS due to mitochondrial harm, is a major contributor to the development of AD, as it facilitates the accumulation and storage of these two proteins [69].

Mitochondrial dysfunction in AD can arise from genetic abnormalities or alterations in gene expression impacting mitochondrial function. Studies have demonstrated that mutations in mtDNA affect respiratory chain function and increase ROS generation [62,67,68], which oxidize mtDNA nucleotides, leading to single-strand breaks. They can also oxidize lipids within the mitochondrial membrane, increasing permeability and promoting the release of pro-apoptotic proteins, such as cytochrome c. Furthermore, ROS-induced mtDNA damage results in the production of 8-hydroxy-2’-deoxyguanosine (8-OHdG) and other oxidized bases [70]. These oxidized nucleotides can cause base-pairing errors during replication and transcription, leading to mutations and replication problems. Single-strand breaks resulting from oxidative damage can further cause the loss of genetic information and impede mtDNA transcription [71].

Altered signaling pathways and the reduced expression of critical transcription factors, due the accumulation of damaged mtDNA, contribute to impaired mitochondrial biogenesis in AD [51,72]. Mitochondrial biogenesis is tightly regulated by several transcription factors and co-activators, including PGC-1α, NRF-1/2, and TFAM, and the reduced expression of these key regulators results in dysregulated biogenesis [73]. Studies have reported reduced PGC-1α expression in the brains of AD patients, likely due to modifications in signaling pathways that suppress its expression. For instance, the Aβ peptide has been shown to inhibit PGC-1α production in neuronal cells, and can also activate the NF-κB signaling pathway, further suppressing PGC-1α production and increasing neuronal cell death. In addition to PGC-1α, the NRF-1 and NRF-2 levels are also reduced in the brains of AD patients [74].

Recent research has emphasized the role of mitochondrial dynamics in AD [75]. Evidence suggests that an imbalance in dynamics contributes to the pathogenesis of this disease, characterized by a rise in mitochondrial fission, reduced fusion, and decreased mitophagy [76]. This dysregulation leads to the buildup of damaged mitochondria, exacerbating dysfunction and contributing to significant neuronal loss, reduced brain volume, and a decline in cognitive function [68,77]. Studies suggest that the interaction between Aβ and dynamin-related protein 1 (Drp1) plays a critical role in mitochondrial malfunction and altered dynamics that are characteristic of AD [78]. pTau may interact with Drp1 and increase its expression, effectively promoting mitochondrial division, compromising mtDNA integrity, and impairing synaptic function in neurons [79,80], ultimately contributing to cognitive deficits [78,81]. The progressive loss of mtDNA can also result in an elevated rate of mitochondrial fission and a reduced level of mitophagy in AD [68]. Furthermore, Aβ and pTau-induced oxidative damage results in lower protein levels of PINK1 and Parkin, which inhibits mitophagy, and contributes to the increased accumulation of Aβ and pTau [76]. These two induce the enlargement of the mPTP, disrupting the IMM permeability barrier and resulting in a reduction in the intermembrane potential [82]. Concurrently, cytochrome c and other pro-apoptotic proteins leak into the cytoplasm through the mPTP, triggering apoptosis and cell death.

In summary, oxidative stress-induced damage to the bases and filaments of mtDNA, impaired mitochondrial biogenesis, and transport defects are among the molecular pathways implicated in mtDNA damage in AD [83]. These pathways may cause mitochondrial dysfunction and ultimately may contribute to the development of AD pathogenesis [84].

### 3.2. Parkinson’s Disease

Parkinson’s disease (PD), the second most prevalent neurodegenerative disorder, is characterized by characteristic motor impairments including tremor and bradykinesia. Neuropathologically, PD is marked by the gradual degeneration of dopaminergic neurons within the pars compacta of the substantia nigra (SNpc), accompanied by the accumulation of α-synuclein aggregates called Lewy bodies (LBs) [85,86], which are mainly composed of misfolded and aggregated forms of the α-synuclein protein, which is a presynaptic protein [87]. This neuronal loss results in a significant reduction in dopamine levels in synaptic clefts and a disruption of neuronal circuitry within dopamine target regions, notably the basal ganglia [88].

In PD pathogenesis, mitochondrial dysfunction is implicated via multiple mechanisms. Notably, a reduction in the activity of the mitochondrial ETC complex I was noted in the substantia nigra of PD patients [89]. As evidence of this, the neurotoxin 1-methyl-4-phenyl-1,2,3,6-tetrahydropyridine (MPTP), which specifically targets mitochondrial complex I, induces PD-like phenotypes in animal models [90]. The reduction in complex I activity leads to a rise in oxidative stress within dopaminergic neurons, compromising their integrity and contributing to neurodegeneration. The deficiencies of this complex are frequently associated with decreased ATP levels, ROS overproduction, and calcium-mediated intracellular damage [91]. Moreover, this reduction in ATP can also facilitate the formation and the accumulation of protein aggregates, including LBs containing α-synuclein, which are key factors in PD progression [51,92]. α-Synuclein accumulation in the OMM can disrupt protein import machinery [93]. Furthermore, accumulating evidence indicate that α-synuclein influences mitochondrial dynamics, particularly by impairing mitochondrial fusion [94,95,96,97].

Mitochondrial dynamics, particularly the distribution of mitochondria to synapses, plays an essential role in both maintaining synaptic function and ensuring mitochondrial health [98]. The relevance of the mitochondrial processes of fusion and fission in dendritic spine and synapse formation is well established. The inhibition of fission reduces the mitochondrial content within dendritic spines, and impairs synaptic development, whereas the stimulation of fission promotes synaptogenesis [99].

Studies has shown genetic alterations in genes that code for mitochondrial proteins. Genetic variants correlated with mitochondrial function and dynamics, including PINK-1, DJ-1, Parkin, and LRRK2, have been related to PD [100].

The significance of mitochondrial function in the progression of familial PD is highlighted by genetic mutations in PARK1 (SNCA), PARK2 (Parkin), PARK6 (PINK1), PARK7 (DJ1), and PARK8 (LRRK2) [101,102]. Many of these genes encode proteins involved in mitochondrial quality control (MQC) processes, such as mitophagy [103].

Parkin is encoded by the PARK2 gene, and it is an essential component in regulating mitochondrial movement and stability via the pathway of ubiquitin-dependent protein degradation. Mitochondrial damage triggers the recruitment of Parkin by PINK1, which subsequently regulates mitochondrial dynamics, involving processes including fission and autophagy/mitophagy.

PINK1 deficiency disrupts mitochondrial morphology and impairs biogenesis, autophagy/mitophagy, and mitochondrial Ca^2+^ homeostasis [104,105]. PINK1 phosphorylates Drp1, a mitophagy regulator, and this phosphorylation is diminished in PD. Parkin interacts with Drp1 and promotes its degradation. Mutations or reduced Parkin levels lead to a higher rate of mitochondrial fission [106,107]. The development of PD is also associated with alterations in mitochondrial biogenesis [108]. SIRT1 expression is decreased in PD, and SIRT1 plays a role in maintaining levels of PGC-1α, a key regulator of biogenesis. PGC-1α activation increases the levels of ROS-detoxifying enzymes, suggesting that stimulating an anti-ROS program could mitigate cellular damage associated with mitochondrial malfunction [109,110].

### 3.3. Amyotrophic Lateral Sclerosis

Amyotrophic lateral sclerosis (ALS) is a progressive and deadly neurodegenerative disease, classifying as the third most common disorder following Alzheimer’s and Parkinson’s diseases [111]. Characterized by the progressive degeneration of motor neurons in the brain and spinal cord, ALS manifests as muscle weakness, atrophy, spasticity, and culminates in death [112]. The disease presents in both sporadic (90–95% of cases) and hereditary forms, the latter arising from genetic mutations. Mitochondrial dysfunction, frequently stemming from protein aggregation and mutations in mitochondrial protein components [113], is a key pathological driver of motor neuron degeneration [60].

Multiple gene mutations are involved in this pathology, including alterations in genes encoding key proteins such as superoxide dismutase 1 (SOD1) [114,115], C9 protein (C9ORF72) [116], CHCHD10 protein, TDP-43 protein (TARDBP) [117], FUS protein (fused in sarcoma/translocated in liposarcoma or heterogeneous nuclear ribonucleoprotein P2), and optineurin (OPTN) [118]. Mutations in these genes increase mitochondrial ROS production and consequently oxidative stress, contributing to neurodegeneration [119,120]. The cytosolic accumulation of TDP-43 aggregates, whether endogenous or exogenous, is related with increased intracellular ROS production, highlighting this protein’s critical role in ALS pathogenesis [121].

SOD1, a multifunctional enzyme, is involved in superoxide radical scavenging, the modulation of cellular respiration, energy metabolism, and post-translational modifications [122]. SOD1 mutations disrupt protein folding, resulting in the buildup of misfolded proteins within the mitochondrial intermembrane space and the formation of free radicals, which ultimately leads to cellular damage and death [123]. ALS pathogenesis linked to SOD1 mutations is primarily caused by mitochondrial dysfunction [124]. Additional research in yeast models has shown that mutant SOD1 can interfere with amino acid biosynthesis and accelerate cell death, which provides further evidence of its involvement in neurodegeneration [125]. Furthermore, studies in yeast models indicate that mutant SOD1 can disrupt amino acid biosynthesis and promote cell death, further supporting its contribution to neurodegeneration [126].

TDP-43, a DNA-binding protein, is a major component of the protein aggregates commonly observed in post-mortem tissue from ALS patients [124]. Mutations within the TDP-43 gene have been found in 3% of familial and 1–15% of sporadic ALS cases, implicating that this protein plays a crucial role in disease initiation [127,128]. These TDP-43 aggregates are typically found within the neuronal cytoplasm of ALS patients, where they disrupt the regulation of mitochondrial transcripts [113,129].

The mitochondrial protein CHCHD10 is essential for the formation and maintenance of mitochondrial cristae junctions. Intramitochondrially, CHCHD10 interacts with CHCHD2, forming a complex vital for efficient mitochondrial respiration. Mutations in CHCHD10 can result in mitochondrial network fragmentation and a loss of cristae junctions. Under mitochondrial stress, CHCHD10 translocates from the mitochondria to the cytoplasm, where it interacts with TDP-43, facilitating TDP-43’s return to the nucleus and preventing aggregate formation. Mutations in CHCHD10 can impair this crucial translocation process [130].

Disruptions in mitochondrial movement are observed in ALS cases. The elevated levels of Drp1 protein are linked with increased mitochondrial fission in models of ALS [131]. Changes in the levels of other proteins regulating mitochondrial dynamics, including Fis1, Mfn1, and OPA1, precede motor neuron loss and symptom onset in SOD1-mutant mouse models [18]. Mitochondrial fragmentation is elevated in both TDP-43- and FUS-mutant models, suggesting a role of these genes in regulating mitochondrial dynamics [132,133].

### 3.4. Multiple Sclerosis

Multiple sclerosis (MS) is a chronic inflammatory neurological disease characterized by multifocal demyelination and progressive neurodegeneration, accompanied by an autoimmune response to autoantigens [134]. This pathology arises from progressive demyelination and axonal dysfunction, resulting in a broad range of symptoms, such as visual and sensory disturbances, ataxia, weakness, sexual dysfunction, depression, pain syndromes, and cognitive deficits [135].

A growing number of studies have identified mitochondrial dysfunction in MS, including increased mitochondrial DNA mutations, altered gene expression, impaired enzyme activity, diminished mtDNA repair capacity, dysregulation of dynamics, and abnormalities in cellular energy metabolism metabolites [13,136]. The confluence of these aberrations culminates in overt mitochondrial dysfunction in MS [136], and neurons, dependent of mitochondrial function, are particularly vulnerable to this dysfunction.

Inflammation, a key pathogenic mechanism in MS, begins with the infiltration of self-reactive lymphocytes into the blood–brain barrier, triggering chronic neuroinflammation. Oxidative stress, resulting from the overproduction of ROS at the inflammatory site, is a critical driver of inflammation exacerbation and in the pathogenesis of MS.

Myelin phagocytosis by macrophages and microglia generates ROS [137], and demyelinating MS lesions exhibit markers of lipid peroxidation, protein oxidation, and 8-hydroxydeoxyguanosine, indicative of DNA oxidative damage [136]. Oxidative mtDNA damage and impaired mitochondrial complex I and IV activity in MS lesions lead to mitochondrial dysfunction, reduced oxidative phosphorylation, and increased ROS generation [136]. This endogenous oxidative stress spreads within neurons and glial cells, damaging intracellular proteins, lipids, and DNA, leading to the generation of secondary metabolites that can function as additional autoantigens. Furthermore, ROS directly injure the myelin sheath, promoting the release of further autoantigens, thus exacerbating autoimmune inflammation and subsequent damage to neuronal structures [138,139].

Oligodendrocytes (OLs), responsible for myelin sheath formation, play a critical role in MS progression. They are especially vulnerable to oxidative damage due to their low antioxidant capacity, making them a prime target for ROS-mediated injury and subsequent axonal demyelination [136]. Recent studies have demonstrated that impairments in the mitochondrial pathways are critical determinants of OL survival and differentiation, both in vitro and in vivo [140].

Mitochondria migrate along axons to demyelinated sites in an attempt to promote the regrowth of myelin through the Miro1 and PGC-1α/PPARγ pathways. However, chronic immune activation compromises this repair mechanism [141]. Furthermore, the inhibition of mitochondrial biogenesis increases the occurrence of dorsal root ganglion neurons that lack myelin markers, promoting the establishment of demyelinated tracts and consequently contributing to axonal dysfunction [142].

Mitochondrial dysfunction, beyond ROS production, disrupts oxidative phosphorylation and other metabolic processes, leading to an imbalance of neurotrophic factors for neurons and oligodendrocytes, resulting in increased axonal demyelination [143]. N-acetylaspartate (NAA), a mitochondrial metabolite and an indirect substrate for oligodendrocyte myelin production, is underutilized by dysfunctional mitochondria, leading to lower acetate levels in the parietal and motor cortices of post-mortem MS tissue samples [144]. Another consequence of neuronal mitochondrial dysfunction is excitotoxicity, arising from impaired neurotransmitter metabolism, which compromises neuronal function and triggers apoptosis [143]. 

Reduced energy efficiency, impaired ionic homeostasis (especially calcium), and the activation of cell death mechanisms are other consequences of mitochondrial dysfunction [143,145,146].

Mitochondrial dynamics, essential for cell viability, are also disturbed in MS. Axonal mitochondrial transport both anterograde and retrograde is inhibited, leading to energy deficits and the accumulation of damaged mitochondria [147,148].

The disruption of anterograde transport prevents the delivery of newly synthesized mitochondria from the neuronal soma to the axon, further exacerbating axonal energy deficits. Conversely, impaired retrograde transport causes the accumulation of damaged and dysfunctional mitochondria within the axon, serving as an additional source of ROS and, consequently, oxidative stress. Concurrently, mitophagy levels are elevated in MS neurons, although it remains unclear whether this process contributes to pathogenesis or represents a compensatory mechanism [149].

Alternative mechanisms like sirtuin dysregulation and Rab32-mediated endoplasmic reticulum stress demonstrate how mitochondrial malfunction acts as a triggering event in progressive neuronal cell death and the exacerbation of this disease [150,151]. Therefore, MS-associated axonal degeneration, resulting from mitochondrial structural and functional deficits, can be viewed as a multi-stage process.

## 4. Targeting Mitochondrial Dysfunction in NDs: Nutraceutical Compounds

The primary limitation of current therapeutic agents for neuronal damage is the high incidence of adverse effects associated with their chronic use. Consequently, there is substantial interest in exploring alternative strategies, including nutraceutical-based therapies.

### 4.1. Nutraceuticals: A Brief Overview

Nutraceuticals are employed in the management and prevention of diseases spanning a spectrum from mild disorders to highly toxic neoplasms [6]. The term “nutraceutical”, coined by Stephen L. DeFelice, is a blend of “pharmaceutical” and “nutrition”. Nutraceuticals, also referred to as phytochemicals or functional foods, are naturally occurring bioactive compounds that confer health benefits by aiding in the treatment and/or prevention of disease [152,153].

Nutraceuticals have more recently been redefined as “food products or their secondary metabolites that provide health benefits (for the treatment and/or prevention of a specific disease) in clinical settings” [154,155]. These health-promoting properties, coupled with their therapeutic potential and favorable safety profile, make them attractive candidates for long-term consumption, as required for the management of chronic conditions such as neurodegenerative disorders [156].

Numerous studies have demonstrated the neuroprotective effects of nutraceuticals (as a form of complementary medicine) against various neurodegenerative disorders. These effects are mediated by the regulation of energy metabolism, neuro-oxidative stress, neuroinflammation, enhanced neurogenesis, and the stabilization of mitochondrial function [157,158,159] via diverse signaling pathways [22,160].

Despite their therapeutic potential, one of the primary challenges that nutraceuticals face is their low bioavailability. This is very much determined by their physico-chemical characteristics such as physical state, chemical structure, solubility, and lipophilicity. Both molecular weight and chemical structure are crucial for determining permeability, intestinal absorption, and lipophilicity and solubility [161,162]. Solubility and lipophilicity represent a compromise in the case of high bioavailability. Hydrophilic biomolecules do not pass through the cell membrane, whereas lipophilic biomolecules are not soluble in gastrointestinal fluids [163,164]. This results in various profiles: some molecules (e.g., EGCG, vitamin C) are very soluble but have low permeability, others (curcumin) have low solubility and permeability, and others (resveratrol) have low solubility but high permeability. Polyphenols are a good case in point of this association: their polymeric or glycosylated forms require enzymatic or microbial hydrolysis for absorption [165,166]. Finally, an interaction with food components of nutraceuticals can result in their degradation and, thus, reduced bioavailability. Fatty acids such as PUFA, CLA, and retinol tend to oxidize [167,168,169], and the presence of antinutrients (for example, phytate) may restrict mineral absorption [170,171]. Interference with food proteins may also regulate the availability of nutraceuticals [172].

### 4.2. Nutraceuticals in ND: A Multi-Targeted Pathway

Nutraceuticals offer a potentially improved and safer integrative approach for neurodegenerative diseases (NDs) due to their reduced side effects and compatibility with conventional pharmacological therapies [5,6,173].

These compounds exert their effects through diverse mechanisms, including antioxidant activity (neutralizing reactive oxygen species/free radicals), anti-inflammatory, anti-excitotoxic, and anti-apoptotic actions, caspase inhibition, modulation of cell signaling pathways, metal chelation, and regulation of mitochondrial homeostasis.

Importantly, many nutraceuticals do not act through a single mechanism but rather via multiple pathways (Table 1). For example, coenzyme Q10, astaxanthin, resveratrol, curcumin, isothiocyanates, and α-lipoic acid have demonstrated therapeutic efficacy against numerous NDs [1,2,5,6]. Furthermore, a substantial body of research suggests beneficial effects of nutraceuticals in various NDs [174]. This is due to the fact that neurons are particularly vulnerable to mitochondrial dysfunction and damage. Consequently, considerable research has focused on the ability of various nutraceuticals to preserve mitochondrial function and thereby confer protection against NDs.

### 4.3. Stimulating Mitochondrial Biogenesis

Several evidence indicates that many nutraceuticals can influence mitochondrial gene expression and induce mitogenesis, the process of mitochondrial biogenesis (Figure 3).

For instance, ursolic acid, a triterpenoid prevalent in various fruits and vegetables, increases mitochondrial mass and promotes ATP production while concurrently reducing mitochondrial ROS generation. This reduction in ROS subsequently activates the redox-sensitive adenosine monophosphate-activated protein kinase (AMPK) pathway and PGC-1α. The activation of the AMPK/PGC-1α pathway leads to the enhanced expression of cytochrome c oxidase (COX) and uncoupling protein 3 (UCP3). Through the activation of these AMPK and PGC-1α pathways, ursolic acid leads to the induction of mitochondrial biogenesis [175,176]. Similarly, ginger extract and its primary constituents, 6-gingerol and 6-shogaol, facilitate biogenesis in mice by enhancing the expression of proteins involved in the oxidative phosphorylation system (OXPHOS) system and activating the AMPK-PGC1α pathway [177].

Curcumin, a major component of turmeric and curry, is derived from the rhizome of *Curcuma longa* [198], and this compound exhibits significant health benefits, including anti-inflammatory and antioxidant properties, and has demonstrated neuroprotective activity [199]. Specifically, curcumin has been shown to ameliorate 6-hydroxydopamine-induced neurotoxicity in MES23.5 cells by partially restoring the ΔΨm, increasing the level of Cu-Zn SOD, and suppressing an increase in intracellular ROS and the translocation of NF-κB, a transcription-factor involved in inflammation [160]. Furthermore, curcumin has been shown to normalize mitochondrial DNA levels and restore mitochondrial oxidative metabolism and biogenesis [178]. Specifically, it upregulates cellular signaling pathways that govern mitochondrial biogenesis, including PGC1α, NRF1, and TFAM.

Polyphenols, beyond their established antioxidant, anti-inflammatory, and cardioprotective effects, are also recognized for their ability to affect the molecular processes involved in the formation of mitochondria [182,200]. Several polyphenols, such as resveratrol [179], hydroxytyrosol [180], and quercetin [181], have been shown to enhance mitochondrial biogenesis by increasing the expression and activity of the transcriptional co-activators SIRT1 and PGC-1α. SIRT1 interacts directly with PGC-1α and deacetylates it, thereby increasing PGC-1α activity. Consequently, polyphenols can stimulate the mitochondrial biogenesis by activating the SIRT1/PGC-1α pathway [182].

Pterostilbene, a stilbenoid with structural similarities to resveratrol, exhibits improved bioavailability [201]. Found in blueberries, grape leaves, and grapevines [202,203], this compound possesses a range of pharmacologically beneficial properties, including anti-inflammatory, anti-apoptotic, antioxidant, anticancer, antidiabetic, cardioprotective, neuroprotective, and anti-atherosclerotic activities. Numerous studies have documented its positive impact on mitochondrial function, encompassing biogenesis, oxidative stress modulation, and apoptosis regulation. Studies suggests that pterostilbene may even surpass resveratrol in its capacity to support mitochondrial biogenesis and regulate mitochondrial redox homeostasis [204].

### 4.4. Regulating Mitochondrial Fusion and Fission

As discussed in previous sections, mitochondrial fusion and fission are dynamic processes crucial for remodeling the cristae of the IMM. These processes play vital roles in mitochondrial transport, ATP production via oxidative phosphorylation, regulation of mitochondrial number and distribution, maintenance of efficient mitochondrial function, and overall homeostasis [205,206]. The dysregulation of these dynamic processes is strongly implicated in neurodegeneration.

Certain nutraceutical compounds are promising for treating neurodegeneration arising from disruptions in mitochondrial dynamics. One such compound is sulforaphane, abundant in cruciferous vegetables, that can modulate the kinetics of mitochondrial fusion and fission by inhibiting histone deacetylases (HDACs) and DNA methyltransferases, thereby preventing both genetic and epigenetic mutations [183].

Furthermore, diosgenin, found in various plants, has demonstrated protective effects by mitigating disruptions in mitochondrial dynamics. This protection is achieved through an increase in the expression of proteins involved in both mitochondrial fusion and fission, including DRP1 and MFN2 [184].

### 4.5. Preventing Mitochondrial Oxidative Stress

Mitochondria are central to ROS metabolism, and the mitochondrial oxidative respiratory system hosts numerous oxidoreductases that facilitate the transfer of individual electrons to oxygen, utilizing potential energy to generate superoxide, a form of ROS. When mitochondrial structure or function is compromised, and the antioxidant defense system is overwhelmed, ROS production disrupts the redox balance [207]. Mitochondrial oxidative stress is a key factor in the development of neurodegenerative diseases [67]. Indeed, the development of major neurodegenerative disorders is closely linked to ROS generation and mitochondrial dysfunction induced by oxidative stress. Consequently, improving mitochondrial antioxidant capacity to preserve mitochondrial homeostasis represents a promising therapeutic strategy. Recent investigations have demonstrated that various nutraceutical compounds can effectively inhibit mitochondrial oxidative stress, offering potential for delaying or treating NDs [208]. For instance, certain medicinal plants and their bioactive constituents can elevate the brain levels of SOD, glutathione (GSH), and catalase (CAT), thereby exerting neuroprotective effects [209]. Natural compounds derived from the medicinal plant *Centella asiatica*, such as asiatic acid, asiaticoside, madecassic acid, and madecassoside [185], are able to protect the I complex in the OXPHOS system and consequently the mitochondrial function. They also have a strong antioxidant capacity as they are able to induce NrF2-related factors to activate the elements of antioxidant response (AREs) to maintain mitochondrial redox balance and activity [186]. Activating the Nrf2/ARE pathway presents a potentially beneficial clinical strategy to improve mitochondrial function in NDs [210,211]. In the PD models, it has been seen that natural plant-derived components offer protection against the neurotoxicity induced by 6-hydroxydopamine. For instance, carnosic acid induced the expression of the catalytic subunits of γ-glutamate-cysteine ligase, superoxide dismutase, and glutathione reductase by reducing GSH [187].

L-sulforaphane, an isothiocyanate abundant in cruciferous vegetables like broccoli, has shown the marked inhibition of dopamine quinone-induced neuronal death. This protection occurs by reducing ROS accumulation, lessening membrane damage, and preventing DNA fragmentation in dopaminergic cell lines (Cath. and SK-N-BE (2)C) and mesencephalic dopaminergic neurons. Furthermore, L-sulforaphane, along with tert-butylhydroquinone, shields neurons and astrocytes from H_2_O_2_-induced oxidative stress in mixed neuron–astrocyte cultures. This protective effect is mediated by stimulating the Nrf2-antioxidant response element transcriptional pathway. Activating the Nrf2 pathway, in turn, promotes the transcription of various antioxidant genes, including γ-glutamate-cysteine ligase (crucial for GSH synthesis), hemeoxygenase, and NAD(P)H:quinone reductase [212,213].

Beyond these, a blueberry-rich diet is recognized for its neuroprotective capabilities, which involve modulating ROS signaling via the CREB and MAP kinase pathways [214]. The neuroprotective actions of resveratrol are believed to stem from its antioxidant properties, its capacity to modulate Aβ processing, and its ability to upregulate sirtuin1, a gene linked to longevity [215]. Both carnosic acid and rosmarinic acid have exhibited neuroprotective effects both in vitro and in vivo by scavenging ROS [216,217]. Aged garlic extract offers protection to PC12 cells against Aβ peptide-induced apoptosis, achieved by suppressing ROS generation and attenuating caspase-3 activation, DNA fragmentation, and PARP cleavage [218]. Eugenol, a nutraceutical derived from cloves, has been shown to prevent 6-hydroxydopamine-induced reductions in dopamine levels in the mouse striatum by decreasing 6-hydroxydopamine-induced lipid peroxidation and elevating GSH levels [219]. Epidemiological studies also suggest that diets rich in antioxidant vitamin supplements like vitamin C and vitamin E are associated with a reduced risk of developing PD [220].

### 4.6. Modulation of Mitochondrial Calcium (Ca^2+^) Homeostasis

The presence of a rapid and specific Ca^2+^ uniport channel in the IMM highlights the essential function of mitochondria in Ca^2+^ homeostasis [221]. Flavonoids exert significant effects on the regulation of calcium-activated mitochondrial high-conductance potassium (mitoKCa) channels [188]. These effects contribute to the inhibition of apoptosis and the promotion of neuronal survival and differentiation, processes that are dependent of the modulation of Ca^2+^ signaling during mitophagy [222]. Beyond flavonoids and their analogs, other natural compounds, including quinic acid and resveratrol, can activate mitochondrial ATP synthase-dependent respiration via increasing mitochondrial Ca^2+^ levels [189,223].

Mitochondria play a crucial role in apoptosis, a process positively regulated by the rate of Ca^2+^ efflux from the cell membrane. Numerous studies have demonstrated that Ca^2+^ efflux is modulated by the key gene Bcl-2, which inhibits pore formation in the mitochondrial membrane, thereby preventing Ca^2+^ efflux [224]. The inhibition of Bcl-2 accelerates calcium efflux, inducing apoptosis. Consequently, excessive Ca^2+^ release acts as a pro-apoptotic signal, triggering a cascade of events that culminate in cell death.

Several plant-derived polyphenols, such as astaxanthin, phenolic acids, coumarin, and lignans, have been shown to exert anti-apoptotic effects on neuronal cells [190,191,192,193]. These compounds function as anti-apoptotic agents through diverse mechanisms, including the upregulation of Bcl-2 and Bcl-xL, the downregulation of Bax and Bak, and the maintenance of Ca^2+^ homeostasis [194,195].

### 4.7. Maintenance of Mitochondrial Membrane Potential (ΔΨm)

A primary contributor to NDs is the disruption of mitochondrial function in nerve cells. Compromised mitochondrial membrane integrity leads to the dissipation of transmembrane potential and an imbalance in membrane potential (ΔΨm), significantly impairing ATP production capacity and compromising mitochondrial membrane selective permeability [225].

Various natural compounds can modulate mitochondrial membrane potential, thereby inhibiting apoptosis [226]. Polyphenols, such as morin and mangiferin, have been shown to preserve mitochondrial membrane potential and prevent caspase activation in neurons, thus inhibiting apoptosis [195]. In a mouse model of central neuronal mitochondrial damage induced by 15 weeks of chronic ethanol exposure, quercetin supplementation significantly mitigated the loss of mitochondrial membrane potential in neurons [227].

### 4.8. Maintenance of Mitochondrial DNA (mtDNA) Stability

Mutations in mitochondrial DNA-encoding enzymes involved in antioxidant systems represent a significant contributor to neurodegeneration. The release of specific mitochondrial proteins is implicated in neuronal cell apoptosis and senescence. Furthermore, proteins encoded by mtDNA play a pivotal role in the pathogenesis of NDs.

Nutraceutical compounds can modulate these mechanisms, offering potential therapeutic interventions. For instance, ginsenoside Rg1, a tetracyclic triterpenoid saponin isolated from ginseng, has been shown to restore mitochondrial activity in neurons by reversing Bax mRNA and protein overexpression and restoring Bcl-2 protein expression [196]. Black tea extract has been shown to promote mtDNA transcription and translation, as well as tyrosine hydroxylase (TH) protein levels and mRNA expression [197].

### 4.9. Human Clinical Studies

While extensive preclinical evidence, encompassing both in vitro and in vivo studies, strongly suggests the potential of nutraceuticals to modulate mitochondrial dysfunction in NDs, the direct translation of these findings to human efficacy remains an area with limited certainty. Human clinical trials are therefore a critical step in rigorously assessing the efficacy, safety, and clinical relevance of these compounds.

Current clinical research has begun to explore specific nutraceuticals in individuals with NDs (Table 2). However, many of these studies are characterized by small sample sizes, variable treatment durations, and often heterogeneous methodologies, which can limit the generalizability and robustness of their findings.

Among the most promising candidates, nicotinamide and MIB-626 (a nicotinamide precursor) are currently undergoing clinical trials [228]. The proposed mechanism of action for nicotinamide involves its conversion to NAD^+^, which then inhibits sirtuin and reduces levels of Thr231-Phosphotau [229]. Preclinical studies in mouse models have provided support for this, showing that the oral administration of nicotinamide can mitigate both Aβ and tau pathologies and improve cognitive function [230].

Another therapeutic approach is the activation of the Nrf2 pathway, leading to the transcription of antioxidant genes and preventing oxidative stress. *Centella asiatica* extract, known for its ability to activate the Nrf2 pathway, is currently the subject of a Phase I clinical trial [231].

Glutathione, involved in the regulation of brain homeostasis and metabolism, is a potent antioxidant and its deficiency or impaired function can directly contribute to the loss of brain neurons [232]. A Phase I clinical trial is evaluating the participants’ dietary supplementation with glycine and N-acetylcysteine, the precursors of glutathione [233].

In the context of ALS, a clinical trial is examining dietary interventions, assessing the efficacy and tolerability of β-hydroxybutyrate ester, a ketogenic body, when administered as a food supplement [234].

For PD, while traditional treatments primarily focus on increasing dopamine levels within the brain, namely, the areas of the midbrain, they generally lack neuroprotective effects. Coenzyme Q10 (CoQ10), a fat-soluble compound and electron transporter within the mitochondrial electron transport chain, acts as a potent antioxidant in its reduced form (ubiquinol) within mitochondrial membranes. While preclinical studies in mouse models have indicated protective effects on dopaminergic neurons [235] and neuroprotective activity during iron-induced stress in these neurons [236], its overall efficacy in human PD remains a subject of debate. It has been hypothesized that CoQ10 treatment might ameliorate mitochondrial defects in PD patients, thereby slowing the progressive decline in motor functions [237]. A meta-analysis by Liu et al. reported that CoQ10 was well tolerated by PD patients and showed increases in activities of daily living (ADLs) compared to placebo therapy [238]. However, more recent investigations, including a Phase III clinical trial, have yielded controversial evidence regarding the beneficial effects of CoQ10 for PD treatment [239,240]. Another meta-analysis further concluded that, while CoQ10 was well tolerated and safe in PD patients, it did not demonstrate superiority over the placebo in terms of motor symptoms [241].

## 5. Conclusions and Future Perspectives

Mitochondria, being highly dynamic organelles, adapt their morphology, number, and function in response to different stress factors and physiological conditions, whereby they are the critical determinants of cell survival and death. Their morphology is regulated by the cyclic processes of fusion and fission, which are essential for mitochondrial movement and division. Mitochondrial dysfunction, due to morphological alterations, mtDNA mutations, and therefore impaired transcription, contributes to the pathogenesis of numerous diseases. In particular, mitochondrial dysfunction plays a crucial role in neurodegenerative diseases. The conventional therapies currently in use have limitations due to the adverse effects associated with their chronic use, so the advent of nutraceuticals has introduced a promising adjunctive therapeutic strategy for dealing with these complex disorders. Nutraceuticals therefore offer a potentially beneficial and complementary approach to conventional therapies, with the aim of mitigating adverse effects by exploiting naturally occurring bioactive compounds. Indeed, recent research has increasingly focused on clarifying the role of these natural compounds as an alternative or complement to conventional drug therapy, targeting one or more relevant pathways.

This review highlights the potential of several nutraceuticals as preventive, integrative, and therapeutic alternatives for the treatment of neurodegenerative diseases. In particular, they have the ability to modulate mitochondrial dysfunction, which is a key process in these diseases, with the aim of preventing pathological outcomes. Given the multifactorial etiology and progressive nature of neurodegenerative disorders, multi-target therapeutic strategies are increasingly recognized as superior to single-drug interventions.

However, the full translation of these promising findings from preclinical research into clinical practice presents several challenges. It is crucial to recognize the inherent limitations of preclinical studies: in vitro and in vivo models, although fundamental for identifying mechanisms of action, often do not fully replicate the complexity, heterogeneity, and chronic progression of human neurodegenerative diseases. Furthermore, differences between species in terms of metabolism, central nervous system physiology, and treatment response may limit the direct transferability of results [242,243,244,245].

A significant additional obstacle is the complex and often fragmented regulatory landscape surrounding nutraceuticals. The regulation of nutraceuticals is a globally ambiguous field, as these products lie at the intersection of foods and drugs. Unlike pharmaceuticals, which are subject to rigorous clinical trials and approval processes before market entry, most nutraceuticals fall under the category of dietary supplements or functional foods. This classification results in a significantly less stringent regulatory framework [246]. For instance, in the United States, the main regulation, the Dietary Supplement Health and Education Act (DSHEA) of 1994, defines dietary supplements and exempts them from pre-market approval by the Food and Drug Administration (FDA). While manufacturers are responsible for product safety and label accuracy, the FDA intervenes only in cases of ascertained safety problems or misleading claims. This “post-market” approach has drawn criticism for its potential lack of oversight on quality and efficacy [247]. In Europe, the situation is more fragmented, with regulations varying among member states and directives at the European Union level. Generally, nutraceuticals are regulated under food legislation, with specific directives for dietary supplements. The European Food Safety Authority (EFSA) plays a crucial role in evaluating health claims and the safety of food ingredients; however, the exact definition and classification of nutraceuticals can differ among countries, creating challenges for market harmonization and for consumers [246,248,249,250,251].

In conclusion, the therapeutic potential of nutraceuticals offers numerous opportunities for further investigation into other natural compounds with potential neuroprotective effects. Future perspectives must prioritize rigorous, large-scale clinical trials to validate their efficacy, safety, and optimal dosages in humans.

## Figures and Tables

**Figure 1 foods-14-02193-f001:**
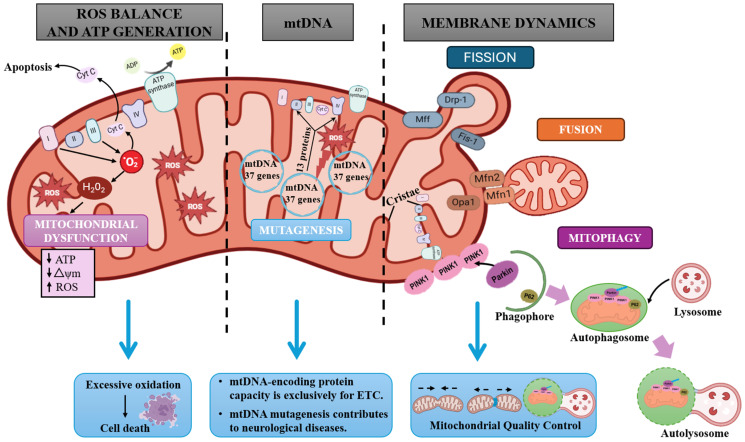
Mitochondrial function and quality control mechanisms: Mitochondria are essential for ATP production and ROS balancing via the electron transport chain (ETC). Their dysfunction can cause reduced ATP production, decreased membrane potential, and increased ROS, leading to oxidation and cell death. The protein-coding information contained within mtDNA is entirely devoted to producing mitochondrial complexes. However, the mitochondrial genome exhibits a significantly elevated mutation rate compared to the nuclear genome. This heightened mutagenesis can severely impair mitochondrial functions, with profound implications for neurological diseases. Mitochondrial membrane dynamics, including mitochondrial fission/fusion, membrane interactions with other organelles, and ultrastructural remodeling of the membrane, is crucial for quality control. The mitophagy system, which includes PINK1 and Parkin, removes damaged mitochondria through the formation of phagophores, autophagosomes, and autolysosomes, preventing the accumulation of dysfunctional organelles. ATP, adenosine triphosphate; cyto *c*, cytochrome *c*; H_2_O_2_, hydrogen peroxide; O_2_^•−^, superoxide; ROS, reactive oxygen species.

**Figure 2 foods-14-02193-f002:**
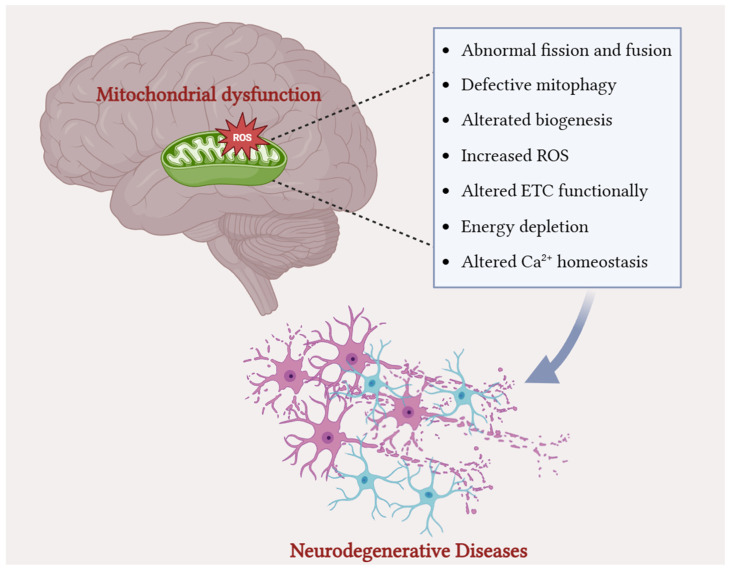
Mitochondrial homeostasis imbalance in neurodegenerative diseases: mitochondrial dysfunction in the brain is a key factor in the development of neurodegenerative diseases. A damaged mitochondrion, represented within brain tissue, is associated with an excessive production of ROS and a series of alterations that compromise mitochondrial homeostasis. Such alterations include abnormal mitochondrial dynamics (fission/fusion imbalance), defective mitophagy, reduced biogenesis, electron transport chain (ETC) dysfunction leading to ATP depletion, and altered calcium homeostasis. These cumulative dysfunctions induce cellular stress and neuronal loss, ultimately culminating in the development of neurodegenerative diseases.

**Figure 3 foods-14-02193-f003:**
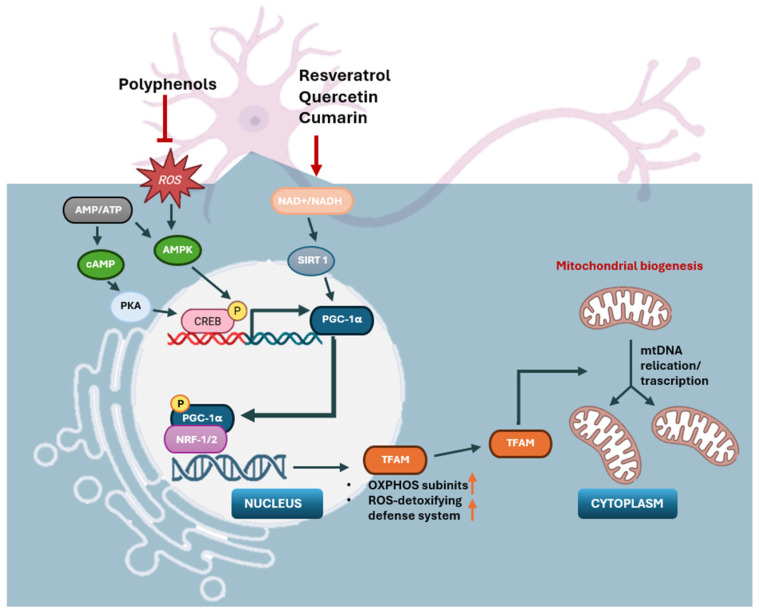
The mechanism of neuronal mitochondrial biogenesis: some nutraceuticals can act on the cascade to improve mitochondrial biogenesis. The mitochondrial activity of neuronal cells depends mainly on the expression of PGC-1α, which is determined by the AMP/ATP ratio. The NAD^+^/NADH report also participates in this way. High concentrations of AMP and Ca^2+^ promote the expression of some kinases and activate PGC-1α through direct and indirect phosphorylation. Subsequently, PGC-1α binds to respiratory nuclear regulatory factors (NRF1/2) to promote the expression of certain proteins in the system, such as mitochondrial transcription factor A (TFAM), which promotes the transcription and translation of mtDNA and ultimately leads to an increase in the growth of mitochondrial biogenesis.

**Table 1 foods-14-02193-t001:** Key nutraceuticals and their mechanisms of action on mitochondrial function relevant to neurodegenerative diseases.

Principal Nutraceutical	Specific Mechanisms of Action on Mitochondrial Function	Refs.
Ursolic Acid	Increases mitochondrial mass and ATP production; reduces mitochondrial ROS generation; activates the AMPK/PGC-1α pathway to induce mitochondrial biogenesis	[175,176]
Ginger Extract (6-gingerol, 6-shogaol)	Facilitates biogenesis in mice by enhancing OXPHOS system protein expression and activating the AMPK-PGC1α pathway	[177]
Curcumin	Normalizes mitochondrial DNA levels; restores mitochondrial oxidative metabolism and biogenesis; upregulates cellular signaling pathways (PGC-1α, NRF1, and TFAM)	[178]
Polyphenols (Resveratrol, Hydroxytyrosol, Quercetin, Morin, Mangiferin)	Enhance mitochondrial biogenesis by increasing expression and activity of the transcriptional co-activators SIRT1 and PGC-1α; preserve mitochondrial membrane potential and prevent caspase activation in neurons, thus inhibiting apoptosis; supplementation significantly mitigated the loss of mitochondrial membrane potential in neurons.	[179,180,181,182]
Sulforaphane	Modulates the kinetics of mitochondrial fusion and fission by inhibiting histone deacetylases (HDACs) and DNA methyltransferases	[183]
Diosgenin	Mitigates disruptions in mitochondrial dynamics by increasing the expression of proteins involved in both mitochondrial fusion and fission (DRP1 and MFN2)	[184]
Compounds from *Centella asiatica* (Asiatic Acid, Asiaticoside, Madecassic Acid, Madecassoside)	Protect Complex I in the OXPHOS system and mitochondrial function; strong antioxidant capacity by inducing NrF2-related factors to activate antioxidant response elements (AREs) to maintain mitochondrial redox balance and activity	[185,186]
Carnosic Acid	Induced the expression of the catalytic subunits of γ-glutamate-cysteine ligase, superoxide dismutase, and glutathione reductase by reducing GSH	[187]
Flavonoids	Exert significant effects on the regulation of calcium-activated mitochondrial high-conductance potassium (mitoKCa) channels	[188]
Quinic Acid	Can activate mitochondrial ATP synthase-dependent respiration via increasing mitochondrial Ca^2+^ levels	[189]
Astaxanthin, Phenolic Acids, Coumarin, Lignans	Exert anti-apoptotic effects on neuronal cells; these compounds function as anti-apoptotic agents through diverse mechanisms, including the upregulation of Bcl-2 and Bcl-xL, the downregulation of Bax and Bak, and the maintenance of Ca^2+^ homeostasis.	[190,191,192,193,194,195]
Ginsenoside Rg1	Restores mitochondrial activity in neurons by reversing Bax mRNA and protein overexpression and restoring Bcl-2 protein expression	[196]
Black Tea Extract	Promotes mtDNA transcription and translation, as well as tyrosine hydroxylase (TH) protein levels and mRNA expression	[197]

**Table 2 foods-14-02193-t002:** Current clinical investigations into mitochondrial dysfunction in neurodegenerative diseases.

ID	Treatment	Phase	N
NCT04430517	NR	I	50
NCT05040321	MIB-626	I/II	80
NCT05591027	*Centella asiatica* product	I	48
NCT04740580	Glycine, NAC	I	52
NCT04820478	Beta hydroxybutyrate ester	N/A	76
NCT00180037	CoQ10	III	696

Abbreviations: N—number of participants; NR—nicotinamide riboside; NAC—N-acetylcysteine; N/A—not applicable.

## Data Availability

No new data were created or analyzed in this study. Data sharing is not applicable to this article.

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
