# Peer review of "Nutraceutical Strategies for Targeting Mitochondrial Dysfunction in Neurodegenerative Diseases"

_foods, 2025, doi:10.3390/foods14132193_

Round 1
Reviewer 1 Report
Comments and Suggestions for Authors
Davi and colleagues have authored a narrative review of nutraceuticals potential to affect mitochondrial dysfunction with relevance for neurodegenerative disorders. I fully agree that this is a very relevant topic. The paper includes a section on mitochondrial biology, followed by a section on mitochondrial dysfunction in NDD, and finally on effects of nutraceuticals on mitochondrial mechanisms relevant to NDD, eg the title topic. While the paper is clear and well written, with a wealth of information and citations (206 references), only 2.5 of the 15 pages are actually covering what the title promises. I would have liked more focus on this aspect. I would also have liked to see the information organized in tables, and a one or two figures outlining how some key nutraceuticals affect NDD-relevant mitochondrial functions- that would have benefited the readers. While all these references are mentioned, there is no information about what type of studies (eg cell studies, animals etc) and no critical consideration. Although it is not promised in the objectives, I would have liked to see a section on human studies, with a critical review of the studies (eg placebo-controlled, sample size and other methodological features).
Reviewer 2 Report
Comments and Suggestions for Authors
The manuscript represents a valuable review that contributes to the advancement of knowledge in the field. It is well written, features excellent figures, and offers a fluid and pleasant reading experience. I congratulate the authors on the outstanding work.
I believe that a few minor adjustments could enhance the scientific rigor and clinical applicability of the review. Therefore, I suggest the following:
-Add a discussion on the limitations of preclinical studies;
-Include information regarding the bioavailability of the nutraceuticals;
-Adopt a more cautious tone in statements regarding clinical efficacy;
-Add a table summarizing the main compounds and their mechanisms of action;
-Expand the discussion on future perspectives and regulatory challenges.
Reviewer 3 Report
Comments and Suggestions for Authors
Based on the title of the manuscript, “Nutraceuticals Strategies for Targeting Mitochondrial Dysfunction in Neurodegenerative Diseases” , I was expecting that this review would cover different aspects of nutraceuticals in the prevention and treatment of neurodegenerative diseases. However, authors did not cover aspects relevant from the food science and nutrition perspectives related to the source of nutraceuticals, their bioavailability, efficacy, and safety. Additionally, the part “Targeting Mitochondrial Dysfunction in NDs: Nutraceutical Compounds” is too short compared to the rest of the manuscript. Therefore, in my opinion, this version of the manuscript is not appropriate for publication in Foods.
Other points:
The Abstract is too general and should contain some more specific data.
In recent years, numerous review papers have been published on this topic. Authors should mention them in the Introduction. What is the novelty and added value of this review compared to others?
Although it is not a systematic review, the authors should mention how the data covered by the manuscript were collected. Were there any inclusion criteria?
Regarding specific nutraceuticals, should distinct data originate from in vitro and in vivo studies?
Discuss the bioavailability of specific nutraceuticals. How can data from in vitro and animal studies be translated into dietary intervention strategies related to neurodegenerative diseases?
Reviewer 4 Report
Comments and Suggestions for Authors
The Review by Federica Davì et al., entitled, “Nutraceuticals Strategies for Targeting Mitochondrial Dysfunction in Neurodegenerative Diseases” the Authors presented a comprehensive survey of recent studies dealing with mitochondrial dysfunction contributes to the genesis and progression of neurodegenerative diseases. The authors discussed recent advances in mitochondrial targeting using nutraceuticals, focusing on their mechanisms of action related to mitochondrial biogenesis, fusion, fission, bioenergetics, oxidative stress, calcium homeostasis, membrane potential and mitochondrial DNA stability. It is an interesting review in Neurodegenerative Diseases. However, the authors should improve the review with minor corrections.
Comments
- The review article needs the graphical abstract.
- In Figure 1, some of the text is not visible. The authors can increase the font sizes for better readability.
- In Figure 1, the text looks like it is stretched, and it can be corrected with the same font throughout the Figure.
- The text should be written in free spaces and not on the Figure component.
- The text should be uniform in font style and size in the Figure.
- The figure legends would benefit from additional detail for the Figures. More comprehensive explanations of each figure’s content, mechanistic approach, etc., will benefit the reader’s interest. The legend for Figure 2 must be improved.
- The authors should include abbreviations section. All abbreviations should be explained at first use. Please examine if all abbreviations (even well-known) are explained at the first use. If you do not so, it makes readers incomprehensible, especially for less advanced readers.
- Minor editing is needed throughout the manuscript, especially in the Discussion.
